# Titanate Nanotubes Engineered with Gold Nanoparticles and Docetaxel to Enhance Radiotherapy on Xenografted Prostate Tumors

**DOI:** 10.3390/cancers11121962

**Published:** 2019-12-06

**Authors:** Alexis Loiseau, Julien Boudon, Alexandra Oudot, Mathieu Moreau, Romain Boidot, Rémi Chassagnon, Nasser Mohamed Saïd, Stéphane Roux, Céline Mirjolet, Nadine Millot

**Affiliations:** 1Laboratoire Interdisciplinaire Carnot de Bourgogne, UMR 6303, CNRS-Université Bourgogne Franche Comté, BP 47870, 21078 Dijon Cedex, France; alexis_loiseau@yahoo.fr (A.L.); remi.chassagnon@u-bourgogne.fr (R.C.); 2Preclinical Imaging Platform, Nuclear Medicine Department, Georges-Francois Leclerc Cancer Center, 21079 Dijon Cedex, France; AOudot@cgfl.fr; 3Institut de Chimie Moléculaire de l’Université Bourgogne, UMR 6302, CNRS-Université Bourgogne Franche Comté, 21078 Dijon Cedex, France; mathieu.moreau@u-bourgogne.fr; 4Department of Biology and Pathology of Tumors, Georges-François Leclerc Cancer Center–UNICANCER, 21079 Dijon Cedex, France; RBoidot@cgfl.fr; 5Institut UTINAM, UMR 6213, CNRS-Université Bourgogne Franche-Comté, 25030 Besançon Cedex, France; said.nasser_mohamed@univ-fcomte.fr (N.M.S.); stephane.roux@univ-fcomte.fr (S.R.); 6INSERM LNC UMR 1231, 21078 Dijon Cedex, France; 7Radiotherapy Department, Georges-Francois Leclerc Cancer Center, 21079 Dijon Cedex, France

**Keywords:** titanate nanotubes, gold nanoparticles, vectorization, nanocarrier, colloidal stability, docetaxel, cytotoxicity, biodistribution, radiotherapy, prostate cancer

## Abstract

Nanohybrids based on titanate nanotubes (TiONts) were developed to fight prostate cancer by intratumoral (IT) injection, and particular attention was paid to their step-by-step synthesis. TiONts were synthesized by a hydrothermal process. To develop the custom-engineered nanohybrids, the surface of TiONts was coated beforehand with a siloxane (APTES), and coupled with both dithiolated diethylenetriaminepentaacetic acid-modified gold nanoparticles (Au@DTDTPA NPs) and a heterobifunctional polymer (PEG_3000_) to significantly improve suspension stability and biocompatibility of TiONts for targeted biomedical applications. The pre-functionalized surface of this scaffold had reactive sites to graft therapeutic agents, such as docetaxel (DTX). This novel combination, aimed at retaining the AuNPs inside the tumor via TiONts, was able to enhance the radiation effect. Nanohybrids have been extensively characterized and were detectable by SPECT/CT imaging through grafted Au@DTDTPA NPs, radiolabeled with ^111^In. In vitro results showed that TiONts-AuNPs-PEG_3000_-DTX had a substantial cytotoxic activity on human PC-3 prostate adenocarcinoma cells, unlike initial nanohybrids without DTX (Au@DTDTPA NPs and TiONts-AuNPs-PEG_3000_). Biodistribution studies demonstrated that these novel nanocarriers, consisting of AuNP- and DTX-grafted TiONts, were retained within the tumor for at least 20 days on mice PC-3 xenografted tumors after IT injection, delaying tumor growth upon irradiation.

## 1. Introduction

Cancer remains one of the world’s most devastating diseases with approximately 0.2% of people worldwide diagnosed with cancer at some point during their life [1]. However, even if mortality has a tendency to decrease, due to a better understanding of tumor biology and improvement of diagnostic tools and treatments, the prevalence of prostate cancer remains very high, especially in developed countries [2]. As an example, in 2019 in the United States, prostate cancer was the third-most diagnosed cancer, with close to 175,000 estimated new cases, corresponding to 1 prostate cancer over 10 detected cancers, being the second leading cause of cancer-related deaths in men with more than 31,000 estimated deaths [3]. In Europe, in 2018, the number of new prostate cancer cases was estimated at 450,000 [4].

Currently, anticancer chemotherapeutic agents such as docetaxel (DTX) are used to target tumor cells in prostate cancer treatment. DTX is an anti-mitotic chemotherapeutic agent, well known to decrease androgen receptor activation in castration-resistant prostate cancer cells [5,6]. It has been approved by Food and Drug Administration (FDA) in particular for the treatment of hormone-refractory prostate cancers [7]. Chemotherapy is often associated with radiotherapy (RT) to increase its efficiency during tumor treatment [8]. Nevertheless, injected drugs weakly reach tumor sites, and patients who undergo repeated treatments develop drug resistance within 24 months of initial exposure [6,9,10]. Thus, high doses, relative to the patient’s needs, are administered, causing harmful side effects and excessive toxicities [11,12,13].

The development of nanotechnologies has offered a new strategy to incorporate and vectorize an active substance specifically to sick cells, increasing its efficacy while limiting systemic concentration. Theranostic nanohybrids have been considerably developed over the past decade as a new generation of nanocarriers for therapeutic and diagnosis purposes [14,15]. This emergent nanotechnology can be used to control injected doses, to perform medical diagnostic imaging based on nanohybrids monitoring inside the organism, and to improve the intracellular concentration of drugs and allow their accumulation within tumor site by enhanced permeability and retention (EPR) mechanisms while limiting toxicity in normal tissues [16,17,18,19]. Nevertheless, further increasing treatment efficacy is a relevant issue. Direct tumor administration through intratumoral (IT) injection might be a relevant approach. Accordingly, there is an interest to develop new nanocarriers for docetaxel to increase therapeutic efficiency and enhance radiation sensitization by maintaining radiosensitizing agents inside cancer cells. Among all nanovectors, carbon nanotubes (CNTs) have been widely investigated for drug delivery applications [20] as well as halloysite clay nanotubes (HNTs) [21]. CNTs are pioneer nanovectors, representing a novel set of nanomaterials available for cancer therapy [22]. Their unique physicochemical properties and their ability to cross cell membranes provide a higher capacity of drug loading when compared to conventional liposomes and dendrimer drug carriers [23]. In addition, CNTs may prolong circulation time and improve bioavailability of conjugated drug molecules [24]. However, these molecules are largely insoluble. To become biologically compatible, appropriate surface modifications leading to a better water solubility should be envisaged [25]. As for HNTs, considered as viable and inexpensive nanoscale containers for encapsulation of drugs [26,27], they also require surface modification for increased colloidal stability, and even though some coated HNTs are evaluated as being non-cytotoxic up to 75 µg.mL^−1^ [27], some others appear to have toxic effects [28]. To overcome CNT and HNT drawbacks, titanate nanotubes (TiONts) have received particular attention as a new generation of nanovectors with adequate surface chemistry that are highly adaptable when compared to the relatively inert chemistry of CNTs. Similarly, TiONts are widely studied [29,30,31] in a broad range of applications since their discovery in the late 1990s [32]. Recently, TiONt applications have been developed in several fields of biomedicine [29,33], such as orthopedics and dental implants [34], dopamine detection [35], DNA transfection [36] and adsorption [37], bioimaging [38,39], safe nanocarrier [36,40,41], drug delivery (genistein and docetaxel) [42,43,44], and cancer cell radiosensitization [44,45]. These TiONt applications are possible due to the atypical morphology shared with CNTs and HNTs. However, they present a shorter length (about 100–300 nm), an opening at the extremities, and a wide versatility of surface chemistry compared with CNTs [29]. It has been shown that shape and functionalization of nanoparticles, used as carriers, affects biodistribution [19,46]. Moreover, our group demonstrated that TiONts can be internalized with no cytotoxicity induction and maintained inside cells for at least 10 days in vitro [36,45]. Finally, the exposure to TiONts combined with irradiation induced a radiosensitizing effect [45]. The functionalization of nanocarrier-TiONts is mandatory in order to have new or complementary functionalities such as stability and biocompatibility in physiological conditions for biomedical applications [19,40,43,47]. In addition, functionalization enables TiONts to carry therapeutic molecules and improves colloidal stability, required for vectorization applications. In very recent studies, our group has reported the use of TiONts as carrier for therapeutic molecules together with DTX (TiONts-DTX) into prostate tumor [43,44]. This nanocarrier was beforehand pre-functionalized with 3-aminopropyl triethoxysilane and with a hetero-bifunctional polymer (polyethylene glycol) to immobilize DTX by covalent linkages. In vivo tests with IT injections of TiONts-DTX showed that more than 70% of TiONt nanovectors were retained within the tumor for at least 7 days. In addition, the radiosensitizing effect of nanohybrids was evaluated on PC-3 tumors with and without RT. In both conditions, tumor growth was significantly slowed down in mice receiving TiONts-DTX compared to mice receiving free DTX. This work proved nanohybrids ability to remain inside the tumor, increasing therapeutic efficiency.

Encouraged by these results, it was necessary to further improve the radiosensitizing effect of these nanohybrids. Over the past decade an increasing interest to use gold nanoparticles (AuNPs) as radiosensitizers for radiation therapy [48,49] has arisen. AuNPs have the ability to combine imaging and therapy on the basis of the strong X-ray absorption cross section due to the high atomic number (Z) of gold [49,50]. They have been commonly used for imaging applications [51,52,53,54] and accumulate in tumors upon the delivery of diagnostic agents and therapeutic drugs, while being effectively excluded from healthy tissue [55,56,57,58]. AuNPs are biologically well-tolerated and present a low toxicity [49,58,59]. Among the numerous examples of gold nanoparticles, Au@DTDTPA NPs appear attractive for image-guided radiotherapy. The ultrasmall gold core confers to the nanoparticles an efficient radiosensitizing effect, which increased the efficiency of radiation therapy by twofold when tumor-bearing animals were treated by radiotherapy after IT injection of Au@DTDTPA NPs [60,61]. Moreover, the organic shell of these nanoparticles has been designed to immobilize gadolinium ions and radioisotopes (indium-111, ^111^In). Consequently, the biodistribution of Au@DTDTPA NPs can be monitored by magnetic resonance imaging (MRI), and by nuclear imaging (positron emission tomography—PET, and single photon emission computed tomography—SPECT) [62,63,64]. Biodistribution studies, performed by a combination of MRI and SPECT, highlighted the safe behavior of Au@DTDTPA NPs after intravenous (IV) injection (accumulation in tumor, no undesirable accumulation in healthy tissue and renal clearance) and provided useful data to determine the ideal delay between the IV injection of Au@DTDTPA NPs and irradiation [60,64]. The radiosensitizing effect of Au@DTDTPA NPs can therefore be better exploited thanks to the possibility of following these nanoparticles by MRI. However, their potential is probably under-exploited due to their fast-renal clearance. The combination of Au@DTDTPA NPs with TiONts is expected to overcome this limitation by (i) the maintaining of nanotubes on site thanks to the design of nanohybrids compared to circulating nanovectors, (ii) improving the efficiency of nanohybrids in IT compared to IV even with the EPR effect, and (iii) the very possibility of combined injection of nanohybrids with radioactive iodine grains during brachytherapy [65]. Moreover, grafting gold nanoparticles onto TiONts, together with anticancer agents, paves the way to associate in the same entity, tumor retention, radiosensitization, and chemotherapy, and seems to be a new, attractive, and versatile platform.

This paper describes and analyzes each step required for the synthesis of a next generation nanohybrid. Analysis was performed using different characterization techniques (scanning transmission electron microscopy (STEM), thermogravimetric analysis (TGA), ζ-potential measurement, X-ray photoelectron spectroscopy (XPS), Fourier-transformed infrared (FTIR), UV-visible and inductively coupled plasma (ICP) spectroscopies) as well as biological tests to evaluate efficacy against prostate cancer. Herein, we also report in vitro bioassays carried out on a human PC-3 prostate adenocarcinoma cells using 3-(4,5-dimethylthiazol-2-yl)-5-(3-carboxymethoxyphenyl)-2-(4-ulfophenyl)-2H-tetrazolium (MTS) assay. In vivo biodistribution assays were performed in PC-3 xenografted prostate tumors, after IT injection, on Balb/c nude male mice. The radiosensitizing efficacy of TiONts-AuNPs-PEG_3000_-DTX nanohybrids in association with RT, on a hormone-independent prostate cancer model, was also studied. Overall, these results describe the impact of a new generation TiONt-based treatment on a model of prostate cancer.

## 2. Results

The final nanohybrid (TiONts-AuNPs-PEG_3000_-DTX) was synthesized step-by-step from titanate nanotubes (TiONts) according to Loiseau et al. [43]. The strategy used to engineer a very versatile platform, which could be used for both nuclear imaging and therapy, is presented in Figure 1, even if the final end product is intended only for therapy and imaging being used only for preclinical developments. In this report, each grafting step has been characterized by different techniques. After hydrothermal synthesis of TiONts, morphological conformity was highlighted by transmission electron microscopy (TEM). As expected, a coiled spiral-shaped structure and an internal cavity, as described in [41,43] and Appendix A is shown. The observed dimensions are in agreement with literature on this compound, showing (10 ± 1) nm in outer diameter, (4 ± 1) nm in inner diameter, and (170 ± 50) nm in length [29,30,31,32,43]. TiONts present a large specific surface area due to their tubular shape ((174 ± 1) m^2^.g^−1^), which is necessary to graft an important number of organic compounds. These ligands improve colloidal stability, dispersion state, and circulation time within the organism.

In a previous work, TiONts, TiONts-APTES, and DTX-PMPI had already been characterized by several analysis techniques [30,39,43]. Briefly, grafting ratio on TiONts surface was determined by thermogravimetric analysis (TGA) (Figure 2). Results are reported in Table 1, and the details of the equations are given in Appendix A.

Then, DTDTPA-modified gold nanoparticles were coupled with TiONts-APTES by peptide bond formation. Peptide coupling was performed between one of DTDTPA’s carboxyl groups present in Au@DTDTPA NPs and an amine function on the TiONts-APTES surface. These Au@DTDTPA NPs are composed of a gold core with a diameter around 2.6 nm and a multilayered organic shell with approximately 120 DTDTPA per nanoparticle according to the literature [52,62,64]. Thanks to TGA, the graft ratio of DTDTPA on AuNPs was evaluated as 5.7 DTDTPA.nm^−2^ (Appendix A). The relative mass loss (Au@DTDTPA NPs) was 50%, which led to molar ratio of 1:2.5 between DTDTPA and AuNPs, respectively. HAADF-STEM (high angle annular dark field scanning transmission electron microscopy) images highlighted the presence of gold nanoparticles on the nanotube surface and nowhere else on the grid, suggesting a good purification of ungrafted Au@DTDTPA NPs (Figure 3a,b). The grafting rate of gold nanoparticles on the TiONt surface was calculated and estimated between 20 and 40 AuNPs/TiONt using several techniques: STEM (via a STEM counting of about several hundred of nanotubes), TGA (the relative mass loss is due to DTDTPA molecules and was 0.40 (±0.05) DTDTPA.nm^−2^ of TiONts; Table 1), and a dosage via ICP (inductively coupled plasma). Then, comparing HAADF-STEM images before (Figure 3c,d) and after (Figure 3e,f) reflected the grafting step of PEG_3000_ on TiONts-AuNPs.

X-ray photoelectron spectroscopy (XPS) analyses were carried out to evaluate the chemical composition of the TiONts-AuNPs and TiONts-AuNPs-PEG_3000_ surface (Figure 4 and Table 2).

FTIR spectroscopy was realized to confirm the effective coatings on TiONts (Figure 5a)—TiONts-APTES with the presence of amino-silane and the corresponding groups (Si-O-Si, C-C-NH_2_, CH_2_); TiONts-AuNPs following the grafting of gold nanoparticles onto TiONts by peptidic coupling ((C=O)-N-H formation as well as (C=O)-OH due to the DTDTPA coating of AuNPs); and finally PEG_3000_ on TiONts-AuNPs-PEG_3000_ with a new C-O_PEG_ specific to the ethylene glycol repeat units. As far as surface modifications of TiONts are concerned, ζ-potential measurements were realized to evaluate changes in terms of isoelectric points (IEP) and potential values at the physiological pH (Figure 5b).

To study the colloidal stability of TiONts nanohybrids in physiological conditions (phosphate buffered saline (PBS) buffer, pH 7.4), tubidimetric studies were realized (Figure 6a) by comparing the decrease in absorbance over time at the wavelength of 600 nm. As expected, initial TiONts and simply modified TiONts-APTES exhibited the lowest stability compared to the further modifications of TiONts including AuNPs, PEG, and DTX, as can be seen also in the picture of the suspension of the final nanohybrids (Figure 6b).

MTS assays on a PC-3 human prostate cancer cell lines were performed to evaluate the cytotoxicity of DTX in the final nanohybrid (Figure 7a). Briefly, an increasing range of DTX (from 0.5 nM to 500 nM) was used to evaluate the cytotoxicity of free DTX and TiONts-AuNPs-PEG_3000_-DTX. For a given DTX concentration, the committed quantity of TiONts-AuNPs-PEG_3000_ (without DTX) (green curve) or Au@DTDTPA NPs (blue curve) corresponded to the quantity present on TiONts-AuNPs-PEG_3000_-DTX (orange curve).

The first in vivo biodistribution images, realized by SPECT coupled with a conventional scanner (SPECT/CT) after IT injection, showed that TiONts-AuNPs-PEG_3000_-DTX rabiolabeled with ^111^In were always retained within tumors six days after IT injection (Figure 7b). Moreover, from 24 h to 6 days, the remaining quantity of nanohybrids in tumors did not seem to decrease and proved the ability of the nanohybrid to maintain DTX and gold nanoparticles within tumors to increase their therapeutic efficacy, as more than 60% of the injected radiolabeled nanohybrids were still inside the tumor after 6 days (Figure 7c) and that there was therefore no release of the radiolabeled Au@DTDTPA (NPs)—they were still grafted to titanate nanotubes. Moreover, gamma counting confirmed these results because more than 42.5 ± 3.7% of nanohybrids were still kept inside tumors 20 days after IT injection (Figure 7d).

Finally, the therapeutic effects on TiONt nanohybrids were investigated by the study of tumor growth delay until a tumor volume of 1000 mm^3^ was reached (Figure 8 and Appendix A) and in the presence or not of radiotherapy (RT). Differences were noted between treatments with or without RT and the final nanohybrids TiONts-AuNPs-PEG_3000_-DTX associated with RT showed (i) the longest delay before tumor growth (over 55 days) and (ii) a significant difference in delay compared to similar nanohybrids without AuNPs (28% higher).

## 3. Discussion

The applied functionalization protocol with 3-aminopropyl triethoxysilane (APTES) tended to limit the multilayered formation of this molecule on the surface of TiONts [66,67]. Thus, the grafting ratio of APTES (2.6 NH_2_.nm^−2^) decreased significantly when compared to results available in the literature [39,43]. Moreover, TiONts-APTES enhanced tube individualization, as compared to bare nanotubes, for a suspension of equal concentration (Appendix A). To some extent, a similar observation can be made on grafted AuNPs that did not seem to stick to each other and were relatively evenly distributed over the whole surface of TiONts.

From XPS analyses, gold was observed on TiONts-AuNPs and TiONts-AuNPs-PEG_3000_, which was consistent with the presence of AuNPs (Table 2 and Appendix A). In addition, other atoms such as nitrogen and silicon corresponding to APTES or even chemical elements identical to those of bare TiONts (Ti, O, and Na) were found. Subsequently, the quantitative analysis revealed an increase in carbon and oxygen content for TiONts-AuNPs, in comparison to TiONts-APTES, consistent with the presence of DTDTPA on nanohybrid. Thereafter, increase of these same chemical elements also showed PEG_3000_ grafting on TiONts-AuNPs-PEG_3000_. A significant decrease for chemical elements such as Ti_2p_, O_1s_, and Na_KLL_ was also observed, according to successive grafting. The decomposition of O_1s_, C_1s_, and N_1s_ peaks in XPS spectra highlighted the formation of secondary amide bonds, characteristic of AuNPs grafting on TiONts via peptide coupling (EDC/NHS) as well as carboxyl functions from DTDTPA (Figure 4a,b). These same peaks also suggested the evolution of components associated to PEG_3000_ grafting via peptide coupling (PyBOP) on the remaining free amine functions, after covalent immobilization of Au@DTDTPA NPs (Figure 4c). Therefore, appearance of two new components was attributed to two types of bonds ((C=O)-NH-C and (C=O)-OH) concerning TiONts-AuNPs and TiONts-AuNPs-PEG_3000_ samples compared to TiONts-APTES, one of which was located at 532.8 eV (7%) for the O_1s_ peak and the other at 288.1 eV (11.2%) for the C_1s_ peak [40,68]. Moreover, the O_1s_ region of TiONts-AuNPs-PEG_3000_ showed an increase for the component located at 532.3 eV, indicating a higher peptide coupling rate and more carboxyl functions (from 7% to 14.3% before and after PEG_3000_ grafting, respectively). Moreover, other components may be also assigned to carboxyl groups ((O=C)-OH), such as those located at ≈531 eV (O_1s_) and ≈286 eV (C_1s_) [68]. These functions were responsible for the increase of these components in comparison to those observed at the same positions for TiONts-APTES—15.3% (O_1s_) and 25% (C_1s_), respectively. These analyses were consistent with what was observed for N_1s_ peak for TiONts-AuNPs and TiONts-AuNPs-PEG_3000_, due to the increase of the 399.8 eV component attributed to ((C=O)-NH-C) groups as well as C-NH_2_ bonds at the expense of C-NH_3_^+^ (401.7 eV) with respect to the N_1S_ region of TiONts-APTES [69]. With regards to TiONts-AuNPs-PEG_3000_, the decomposition of C_1s_ and the O_1s_ threshold highlighted the grafting of the polymer. The component associated with C-C/C-H groups at 284.6 eV was more intense due to the PEGylated chains of PEG_3000_ (from 46.9% to 53.1%). Regarding the contributions of the C-O_PEG_ bond, they may be located at 530.8 eV (O_1s_) and 286.1 eV (C_1s_) [40,69].

FTIR spectroscopy confirmed the XPS results, thus showing the effective synthesis of TiONts-AuNPs and TiONts-AuNPs-PEG_3000_ nanohybrids. Indeed, IR spectra show the appearance of new characteristic vibration bands in Figure 5a, corresponding to the formation of amide bonds located at 1050 and 1550 cm^−1^ ((C=O)-N-H) as well as 1720 cm^−1^ ((C=O)-N-H). Furthermore, IR analysis was consistent in that Au@DTDTPA NPs remained on the surface of TiONts even after last synthesis steps, with the persistence of functions attributed to (C=O)-N-H and (C=O)-OH of DTDTPA molecules (violet highlights on IR spectrum) [62]. This was corroborated by the HAADF-STEM images (Figure 3 and Appendix A). Moreover, the strong absorption band at 1100 cm^−1^ (C-O_PEG_) was also observed (green highlights on IR spectrum) [47]. Finally, functions attributed to aliphatic carbon chains were increasingly intense due to the additional organic moieties after each grafting (APTES, Au@DTDTPA NPs, and PEG_3000_) on TiONts, and were located between 1450 and 1300 cm^−1^.

ζ-potential measurements indicated an isoelectric point (IEP) at pH 6.9 for TiONts-APTES. This value was higher than for bare TiONts (IEP 3.3) due to the coating of amine functions (Figure 5b). Therefore, the decrease in the number of amines partly engaged in Au@DTDTPA NPs coupling with TiONts-APTES and appearance of COO^−^ groups from DTDTPA led to an IEP shifted downward to the lower pH value of 5.1 (violet curve). At pH 7.4, carboxylate functions significantly improved the ζ-potential, in absolute value, favoring the electrostatic repulsion—from −6 mV (TiONts-APTES) to −20 mV (TiONts-AuNPs). Nevertheless, a significant screening effect was observed concerning PEG_3000_-functionalized TiONts (green curve) over the entire pH range studied (−2 mV at pH 7.4). These results were striking when compared to a previous study [43], thus suggesting that the steric effect mainly governs colloidal stability at physiological pH. The electrokinetic monitoring of TiONts-AuNPs-PEG_3000_-DTX (orange curve) showed a less pronounced screening effect (−7 mV at pH 7.4), even if the evolution of ζ-potential measurements as a function of pH were very close to that of nanohybrid without DTX-PMPI. The strong screening effect was observed while the grafting of Au@DTDTPA NPs induced a less important grafting density of PEG_3000_ on the surface of nanohybrid than without Au@DTDTPA NPs (0.04 PEG_3000_.nm^−2^ for TiONts-AuNPs-PEG_3000_ vs. 0.05 PEG_3000_.nm^−2^ for TiONts-PEG_3000_; Table 1) [43] due to the decrease in the number of free amines and steric hindrance. The lower coverage rate of polymer led to an area per chain of 25 nm^2^ (the PEG_3000_ radius of gyration is about 2.5 nm, which corresponded to a covering surface of 20 nm^2^), indicating a mushroom conformation [70]. However, this calculation did not consider the TiONts surface already occupied by gold nanoparticles. Therefore, the polymer could also have been in brush conformation onto nanohybrid.

DTX was modified by *p*-maleimidophenyl isocyanate (PMPI) to obtain an adequate function for the combination with TiONts-AuNPs-PEG_3000_. The reaction between PMPI and thiol groups was expected at pH 7.4 in PBS, as maleimide reacts specifically with thiol in a pH range from 6.5 to 7.5 (Appendix A), whereas the reaction was possible both with thiol and amine functions above a pH of 7.5. The functionalization of TiONts-AuNPs-PEG_3000_ with DTX-PMPI was observed by TGA. The relative mass loss of final nanohybrids was greater than that of TiONts-AuNPs-PEG_3000_, revealing the immobilization of therapeutic agent (DTX) on the surface of TiONts (Figure 2). The grafting ratio was estimated to 0.30 DTX-PMPI.nm^−2^ (Table 1) and higher than previously reported (0.20 DTX-PMPI.nm^−2^) [43]. Indeed, thiol functions brought both by DTDTPA and PEG_3000_ could react with the maleimide group of DTX-PMPI in PBS (0.1 M; pH 7.4). However, despite repeated purifications, it cannot be excluded that DTX-PMPI clung/adsorbed to amine groups of APTES not functionalized by Au@DTDTPA NPs in the cavity of nanotubes and/or was trapped within PEGylated chains (Appendix A). Consequently, this enhancement of DTX quantity could increase the therapeutic effect of nanohybrid on tumor cells in addition to AuNPs.

Functionalization with Au@DTDTPA NPs and PEG_3000_ enhanced tube individualization as compared to bare nanotubes at similar concentration (Figure 3c–f). Even if the graft ratio of PEG_3000_ was low, their presence seemed to limit nanohybrids agglomeration, comparing their dispersion with the STEM images of TiONts-AuNPs. The hypothesis, shown above, concerning the conformation of PEG_3000_ on the surface of TiONts (polymer brush conformation), could then be confirmed.

Colloidal stability of different functionalized TiONt suspensions was also investigated under physiological conditions (PBS 0.1 M; pH 7.4) by turbidimetric analyses (Figure 6a) and correlated with STEM images. Results demonstrated a very good colloidal stability of TiONts-AuNPs and TiONts-AuNPs-PEG_3000_ in comparison with TiONts and TiONts-APTES. Indeed, the measured absorbance did not significantly change after 150 min for AuNP-functionalized TiONts. The grafting of DTX-PMPI on the surface of TiONts-AuNPs-PEG_3000_ also did not lead to any major change on the colloidal stability of the final nanohybrid in PBS. Thus, colloidal stability was dramatically improved at physiological pH and was largely sufficient over time to in vivo inject the nanohybrid after radiolabeling the DTDTPA by ^111^In. Moreover, Figure 6b shows that brown TiONts-AuNPs-PEG_3000_-DTX suspension was always stable after 24 h in PBS, proving the presence of gold nanoparticles (TiONts suspensions are white).

Initial results demonstrated that Au@DTDTPA NPs and TiONts-AuNPs-PEG_3000_ did not exhibit any cytotoxicity in the studied range (either from 3 × 10^−3^ to 3 µg.mL^−1^ of Au@DTDTPA NPs and 4.1 × 10^−3^ to 4.1 µg.mL^−1^ of TiONts-AuNPs-PEG_3000_), whereas unmodified DTX showed cytotoxicity (black curve), with a half-maximum inhibitory concentration (IC_50_) of 3.1 nM, in agreement with results previously shown in literature [43,71]. TiONts-AuNPs-PEG_3000_-DTX were still cytotoxic (IC_50_: 82 nM) even though they were less toxic than free DTX. Nevertheless, the cytotoxic efficacy of the final nanohybrid was higher with gold nanoparticles than what was previously observed for TiONts-DTX (IC_50_ = 360 nM), synthesized by Loiseau et al. [43]. Indeed, these nanohybrids being better dispersed and more stable in suspension have probably improved cell internalization by diffusion or endocytosis processes of current nanohybrids in sick cells [45] and led to a better access of DTX to microtubules. These achievements were very promising and allowed in vivo experiments on PC-3 xenografted tumors into Balb/c nude mice.

Twenty days after IT injection, a small amount of nanohybrids was found in other organs such as liver (2.4 ± 0.9%), kidney (1.2 ± 0.4%), lung (0.10 ± 0.01%), spleen (0.20 ± 0.05%), and bowel (0.90 ± 0.01%). Less than 0.1% of nanohybrids were detected in bladder, blood, and heart. The low quantities detected in different organs were correlated with the lack of toxicity shown during mice follow up, which lasted three months post-injection.

Using this in vivo approach, we observed that tumor volumes revealed a slight growth delay after IT injection in mice. Indeed, TiONts-DTX and TiONts-AuNPs-PEG_3000_-DTX groups without radiotherapy (RT) reached a volume of 1000 mm^3^ at a later time when compared with the control group (40.7 ± 5.4 days, 39 ± 4 days, and 35.8 ± 4.5 days, respectively) (Figure 8). This interesting effect may be explained by the retention of DTX within tumor cells by TiONt-based nanovectors improving therapeutic efficacy and preventing its diffusion throughout the body. Thus, these results are consistent with biodistribution analysis. Moreover, these observations have already been described in prior studies evaluating the efficacy of TiONts-DTX with and without RT, in comparison with groups receiving free DTX [43,44]. More importantly, we observed an improved therapeutic efficacy by combining TiONts-AuNPs-PEG_3000_-DTX with RT. Indeed, tumor growth was significantly slowed by TiONts-AuNPs-PEG_3000_-DTX associated with RT to reach a volume of 1000 mm^3^ (55.2 ± 6.9 days), compared with TiONts-DTX with RT (49.9 ± 2.5 days) in the same conditions (*p* = 0.035). Thus, these results suggest that gold nanoparticles significantly improve the RT efficacy of nanohybrid even if the gold quantity injected (corresponding to 36.1 nmol of Au or 66.5 pmol of AuNPs and 15 µg of Au@DTDTPA/animal) was significantly less than the quantity used in previous publication on Au@DTDTPA NPs alone (160 µg of Au@DTDTPA/animal) [61], thus showing the synergistic effect of the association of TiONts and AuNPs nanohybrids fulfilling their role as carriers by concentrating the therapeutic and chelating agents within cancer cells.

## 4. Materials and Methods

### 4.1. Materials

Titanium dioxide (TiO_2_) rutile precursor was purchased from Tioxide (Calais, France). Sodium hydroxide (NaOH), 3-aminopropyl triethoxysilane (APTES), ethanol, benzotriazole-1-yl-oxytripyrrolidinophosphonium hexafluorophosphate (PyBOP), *N*,*N*-diisopropylethylamine (DIEA), diethylenetriaminepentaacetic acid bis(anhydride) (DTPA-BA), tetrachloroauric acid trihydrate (HAuCl_4_. 3H_2_O), sodium borohydride (NaBH_4_), acetic acid, methanol, dimethyl formamide (DMF), trimethylamine, and aminoethanethiol were acquired from Sigma-Aldrich (Saint-Quentin-Fallavier, France). A derivative of polyethylene glycol, named α-acid, ω-thiol-polyethylene glycol (HS-PEG_3000_-COOH, MW = 3073 g.mol^−1^), was purchased from Iris Biotech GmbH (Marktredwitz, Germany). *N-*hydroxysuccinimide (NHS), 1-ethyl-3-(dimethylaminopropyl) carbodiimide hydrochloride (EDC), *p*-maleimidophenyl isocyanate (PMPI), and tris(2-carboxyethyl)-phosphine hydrochloride (TCEP) were obtained from Thermo Scientific (Illkirch, France). Docetaxel (DTX) was purchased from BIOTREND Chemikalien GmbH (Cologne, Germany). Borate buffered saline was prepared from boric acid (99.8%). Phosphate buffered saline (PBS) 1× solution (Fisher Bioreagents, Illkirch, France), dimethyl sulfoxide (DMSO extra dry, anhydrous 99.99%) (Acroseal), and hydrochloric acid (HCl) were also obtained from Fisher Chemicals (Illkirch, France). Only Milli-Q water (ρ = 18 MΩ cm) was used in the preparation of aqueous solutions and to rinse gold nanoparticles. The ultrafiltration cell (Model 8400, 400 mL) and membranes (regenerated cellulose) were purchased from Merck Millipore (Molsheim, France). Gold nanoparticles were filtered using a 0.22 µm pore diameter polymer membrane purchased from Osmonics Inc (Penang, Malaysia). All chemicals were used without further purification.

### 4.2. Preparation of Bare TiONts and Amine-Functionalized TiONts (TiONts-APTES)

Bare TiONts were synthesized by a hydrothermal method. A total of 1 g of precursor (titanium dioxide rutile) was ultrasonicated (30 min, 375 W, Sonics Vibra-Cells (Newton, CT, USA) in a NaOH aqueous solution (10 M, 250 mL). Subsequently, the mixture was transferred into a Teflon reactor with mechanical stirring and heating at 155 °C for 36 h. TiONts were washed by centrifugation (24,000 × *g* for 10 min), dialysis (Cellu·Sep tubular membranes of 12–14 kDa), and ultrafiltration (regenerated cellulose membranes with a molecular weight cut-off (MWCO) of 100 kDa) [30,31,43]. Subsequently, TiONts were functionalized with silane-coupling agent, presenting high reactivity with hydroxyl groups on the surface of material. Consequently, TiONts were modified with APTES (the molar ratio between hydroxyl functions of TiONts and APTES was 1:3) via hydrolysis and condensation in a solution of water and ethanol (50:50 *v:v*) under magnetic stirring at 60 °C for 5 h (TiONts-APTES) [39,47,67]. After the reaction, suspension was ultrafiltered (100 kDa) to eliminate the excess of APTES. Finally, the TiONts-APTES were freeze-dried.

### 4.3. Synthesis of Dithiolated Diethylenetriaminepentaacetic Acid (DTDTPA) and Functionalized Gold Nanoparticles Synthesis (Au@DTDTPA NPs)

The synthesis of Au@DTDTPA NPs was described by Alric et al. [52,63]. Briefly, 5.6 × 10^−3^ mol of dithiolated diethylenetriaminepentaacetic acid bis(anhydride) (DTPA-BA) was dissolved in DMF and heated to 70 °C. Then, 1.23 × 10^−2^ mol of aminoethanethiol was dissolved in DMF and 1.74 mL of triethylamine. This solution was added and mixture was stirred magnetically at 70 °C overnight. Subsequently, the solution was cooled to 25 °C and placed in an ice bath. A white powder (NEt_3_·HCl) was seen to precipitate out and was filtered. After filtration, chloroform washing, and drying under vacuum, DTDTPA was obtained as a white powder.

Gold nanoparticles were synthesized adapting Brust’s protocol in the presence of DTDTPA to control size and colloidal stability [72]. In a typical preparation of gold nanoparticles, 5.1 × 10^−5^ mol of HAuCl_4_·3H_2_O was dissolved in methanol and mixed with 9.4 × 10^−5^ mol of DTDTPA in water, and acetic acid was added to the gold salt solution while continuously stirring the mixture. After 5 min, 5 × 10^−5^ mol of NaBH_4_ dissolved in water was added to the orange mixture under vigorous stirring at 25 °C for 1 h, before adding HCl solution. After partial removal of the solvent under reduced pressure at a maximum temperature of 40 °C, the precipitate was filtered. The resulting black powder (Au@DTDTPA NPs) was dried and either stored as a solid or dispersed in 10 mL of 0.01 M NaOH solution.

### 4.4. AuNP-Coated TiONt (TiONts-AuNPs) Synthesis

TiONts-APTES were mixed with Au@DTDTPA NPs in 40 mL of phosphate buffered saline (0.1 M; pH 7.4). A large excess of EDC and NHS were added beforehand on water (pH 5), to activate the carboxylate functions of the DTDTPA on the surface of AuNPs, during 90 min under magnetic stirring. The molar ratio between amines on the surface of the TiONts-APTES and the carboxylate functions of the DTDTPA was 1:0.6. The reaction took place under magnetic stirring for 24 h. Then, TiONts-AuNPs were washed by ultrafiltration (500 kDa) and freeze-dried. Elimination of non-grafted AuNPs was optimized by UV-control of washing waters.

### 4.5. Grafting of Polyethylene Glycol (PEG_3000_) on TiONts-AuNPs

Heterobifunctional polymers HS-PEG_3000_-COOH were activated with PyBOP in a molar ratio of 1:1. The reaction took place in DMSO in the presence of the organic base DIEA (excess) under magnetic stirring and nitrogen flow for 30 min. TiONts-AuNPs were dispersed in DMSO and added to activation solution for 24 h under magnetic stirring and nitrogen flow at 25 °C. Polymers were grafted on the remaining amine functions of APTES. The molar ratio was 1:1 between amine functions (initially present on TiONts-APTES even if gold nanoparticles were present) and polymers. Finally, the product (TiONts-AuNPs-PEG_3000_) was washed by centrifugation (20,000 × *g* for 20 min), then purified by ultrafiltration (500 kDa) and freeze-dried.

### 4.6. Activation and Grafting of the Therapeutic Agent: DTX

Activation and grafting of the therapeutic agent have been described by Loiseau et al. [43]. First, DTX and PMPI (DTX-PMP) were dissolved in DMSO and then added in borate buffered saline. The molar ratio was 1:4, respectively, under magnetic stirring at 25 °C for 24 h. The solution of DTX-PMPI was dialyzed (0.5–1 kDa) and freeze-dried to obtain a yellowish powder. TiONts-AuNPs-PEG_3000_-DTX were synthesized from TiONts-AuNPs-PEG_3000_ and DTX-PMPI (large excess) using TCEP in PBS (0.1 M; pH 7.4). The mixture was homogenized beforehand in an ultrasonic bath and placed under magnetic stirring for 24 h at 25 °C. TiONts-AuNPs-PEG_3000_-DTX were dialyzed, ultrafiltered (500 kDa), and freeze-dried.

### 4.7. Characterization Techniques of Nanohybrids

#### 4.7.1. Thermogravimetric Analysis (TGA)

The amount of the molecules on the surface of the TiONts after each grafting step was determined by TGA (TA instrument, Discovery TGA (New Castle, DE, USA)). An air flow rate of 25 mL.min^−1^ and a temperature ramp of 10 °C.min^−1^ from 50 to 800 °C were used for measurements.

#### 4.7.2. Surface Area Measurements

Specific surface area measurements were carried out using a Micromeritics Tristar II apparatus. Samples were outgassed in situ under 20 mTorr pressure for 16 h at 100 °C. Brunauer–Emmett–Teller (BET) method (S_BET_) was used in the calculation of specific surface area value from N_2_ gas adsorption.

#### 4.7.3. Transmission Electron Microscopy (TEM)

Nanotube morphology and agglomeration state characterization were carried out with a JEOL JEM-2100F, operating at an accelerating voltage of 200 kV and fitted with an ultra-high pole-piece achieving a point-to-point resolution of 0.19 nm. HAADF-STEM micrographs of AuNP-loaded TiONts were taken on this instrument, equipped with a field emission gun (FEG) type cathode. Samples were prepared by dropping a dilute suspension of powders onto the carbon-coated copper grids.

#### 4.7.4. Inductively Coupled Plasma (ICP) Spectroscopy

Determination of titanium and gold contents in final nanohybrids was performed by ICP coupled to mass spectrometry (ICP-MS) analysis (ThermoScientific iCAP 6000 series ICP Spectrometer (Waltham, MA, USA)). A total of 2 mg of final nanohybrids were dissolved in aqua regia at 40 °C. The resulting solutions were diluted in HNO_3_ for analysis.

#### 4.7.5. X-ray Photoelectron Spectroscopy (XPS)

XPS measurements were collected with a PHI 5000 Versaprobe apparatus from a monochromatic Al Kα_1_ radiation (EKα_1_ (Al) = 1486.7 eV with a 200 μm diameter spot size, accelerating voltage of 12 kV, and power of 200 W). Powders were deposited on an indium sheet and then pressed. A Shirley background was subtracted and Gauss (70%)–Lorentz (30%) profiles were applied. Data analysis and curve fittings were realized with CasaXPS processing, and MultiPak software was employed for quantitative analysis. Neutralization process was used to minimize charge effects. Titanium 2p peak (458.7 eV) was used as a reference and allowed the correction of charge effects. The resolution was 2.0 eV for global spectra and 1.3 eV for windows corresponding to selected lines.

#### 4.7.6. Fourier Transformed Infrared (FTIR) Spectroscopy

FTIR spectra were recorded on a Bruker Vertex 70v using OPUS version 3.1. using the KBr method, in which the pellets were made by mixing 2 mg of sample within 198 mg of dried KBr.

#### 4.7.7. ζ-Potential Measurements

A Malvern Nano ZS instrument supplied by DTS Nano V7.11 software was used to determine zeta potentials of nanoparticle suspensions. pH titrations were performed using aqueous solutions of HCl (0.1 M), NaOH (0.1 M), or NaOH (0.01 M) to adjust the pH from 3 to 11. Before each measurement, suspension was prepared in aqueous NaCl solution (10^−2^ M) and sonicated for 10 min.

#### 4.7.8. UV-Visible

Shimadzu UV-2550 was used to measure UV-visible absorbance at 600 nm. Turbidimetric studies of nanoparticle suspensions were made in PBS (0.1 M; pH 7.4) at 25 °C (one measurement/5 min).

### 4.8. Radiolabeling with Indium-111

For in vivo biodistribution studies, DTDTPA molecules grafted on the nanohybrid were labeled using indium-111 radionuclide (^111^In radioactivity half-life t_1/2_ = 67.9 h) [73]. The preparation of ^111^In-labelled nanohybrids was performed by adding ^111^In chloride to TiONts-AuNPs-PEG_3000_-DTX in ammonium acetate buffer. Briefly, 386 MBq of ^111^InCl_3_ in 0.05 M HCl (500 µL) were mixed with 50 µL of 1 M AcONH_4_ pH 7.07 and 450 µL of 0.1 M AcONH_4_ pH 5.8, and 2 mg of TiONts-AuNPs-PEG_3000_-DTX was then added. The resulting mixture (pH 5) was stirred overnight (16 h) at 37 °C in a Thermomixer. Instant thin layer chromatography (ITLC) was performed to determine the radiolabeling yield and to assess the absence of free ^111^In. A total of 1 µL of the nanohybrids mixture was spotted on the ITLC-silica gel (SG) strip, which was subsequently eluted with sodium citrate 0.1 M pH 5, and the strip was then analyzed using an AR-2000 radiochromatograph (Eckert and Ziegler, Berlin, Germany; Rf = 0 for radiolabeled nanoparticles whereas Rf = 1 for small ^111^In-AcO). At the end of incubation, suspension was centrifuged (13,000 × *g*, 15 min) and supernatant was discarded. The radiolabeled nanohybrids were then suspended in saline prior to injection.

### 4.9. Cells and Animals

Human PC-3 prostate adenocarcinoma cells (ATCC, Manassas, VA, USA) were cultured in Dulbecco’s modified Eagle medium (DMEM) with 10% fetal serum calf (Dutscher, France) at 37 °C, 5% CO_2_, and 95% humidity.

Two days prior to mice injection with cancer cells, whole-body irradiation was performed with a γ-source (2 Gy, 60Co, BioMep, Bretenières, France). The injection unit included 10 × 106 PC-3 cells in 200 μL serum-free culture medium containing Matrigel (50:50, *V:V*, BD Biosciences). Injection was performed subcutaneously on the right flank of immunosuppressed athymic Balb/c nude male mice which were at least six weeks of age (Charles River, L’Arbresles, France). All mice were housed in our approved animal facility (Centre Georges--François Leclerc, Dijon, France) and all experiments followed the guidelines of the Federation of European Animal Science Associations. All animal studies were conducted in accordance with the European legislation on the use of laboratory animals (directive 2010/63/EU) and approved by accredited ethical committee of the Grand Campus (Dijon, France). The Ministry project agreement numbers are #13968 (for radiotherapy experiments) and #7830 (for imaging experiments), and the ethical committee agreement number is 105, with the official name “C2ea Grand Campus”.

### 4.10. Treatments

Mice were randomized 20 days post cancer cells injection. To distribute mice among the different treatment groups, a randomization was performed. The aim was to obtain an equivalent average tumor volume (TV) in each treatment group (about 200 mm^3^). Before and during irradiation, each mouse was anesthetized with 2.5% isoflurane mixed with oxygen (MINERVE system, Esternay, France).

One hour before the first RT fraction, NPs were delivered intratumorally (50 µL, 1.87 µg/µL for TiONts-DTX and 2 µg/µL for TiONts-AuNPs-PEG_3000_-DTX, in order to have the same DTX concentration in both cases). Radiotherapy was delivered using three daily fractions of 4 Gy by a small animal irradiator (SARRP, Xstrahl, United Kingdom), with 225 kV energy X-ray photons and a dose rate of 3.1 Gy/min. For each RT session, an anterior field and a posterior field were used to irradiate the tumor in a targeted way with a homogeneous dose.

### 4.11. In Vitro Evaluation of Nanohybrid Cytotoxicity

To evaluate the cytotoxic activity of DTX on the surface of nanohybrids, androgen-independent PC-3 prostate cancer cells were seeded in 96-well plates at a concentration of 3000 cells/well and incubated at 37 °C in 190 µL of drug-free culture medium (DMEM) with 10% fetal bovine serum (FBS) for 24 h before treatment (when the cells were at around 20% confluence). Cytotoxicity assays were performed with four samples at each concentration of free DTX (positive control), Au@DTDTPA NPs, TiONts-AuNPs-PEG_3000_, or TiONts-AuNPs-PEG_3000_-DTX. Tumor cells were incubated (+10 µL of drug in 190 µL of culture medium) with a range of equivalent DTX concentrations from 0.5 to 500 nM (100 nM of DTX corresponds to 0.2 µg of nanohybrids per well from TGA, i.e., a nanohybrid concentration of 1.0 µg.mL^−1^). After 96 h of incubation, cell viability was evaluated using MTS assay (Promega Corporation, Madison, WI, USA) according to Mirjolet et al. [44,45,74]. Results were expressed as relative absorption at 490 nm relative to the untreated control.

### 4.12. Analysis of TiONts-AuNPs-PEG_3000_-DTX Biodistribution

Because the organic shell of Au@DTDTPA NPs ensures the immobilization of ^111^In ions (due to the chelating properties of DTDTPA) [64], TiONts-AuNPS-PEG_3000_-DTX can be radiolabeled. The location of these hybrid nanostructures can therefore be monitored by SPECT. After mice in vivo injection, TiONts were tracked using a NanoSPECT/CT small animal imaging tomographic gamma-camera (Bioscan Inc., Poway, CA, USA). TiONts-AuNPs-PEG_3000_-DTX-^111^In (50 μL, 40–60 μg; 5.7–8.7 MBq of activity) were injected into nine subcutaneous PC-3 human prostate tumor-bearing mice. In vivo biodistribution at 1 h, 3 h, 24 h, 48 h, 72 h, and 6 days after injection was analyzed using SPECT/CT imaging. Then, 10 days post-injection, the 3 imaged animals were sacrificed, and 15 days and 20 days after injection the other mice (three animals per group) were also sacrificed. Tumor, blood, lung, liver, kidney, spleen, bladder, bowel, and heart of each mouse were collected and radioactivity in these samples was measured using a gamma counter (Wizard 1480, Perkin Elmer, Waltham, MA, USA).

### 4.13. Evaluation of the Radiotherapeutic Efficacy of TiONts-DTX and TiONts-AuNPs-PEG_3000_-DTX

To evaluate the benefit of nanohybrids, tumors were treated with an intratumoral injection of a 50 μL TiONts-DTX or TiONts-AuNPs-PEG_3000_-DTX suspension (containing 10.5 nmol of DTX grafted onto 93.5 μg of TiONts-DTX and 100 μg of TiONts-AuNPs-PEG_3000_-DTX). The gold quantity injected present on TiONts-AuNPs-PEG_3000_-DTX was 15 μg Au@DTDTPA/animal (corresponding to 36.1 nmol of Au or 66.5 pmol of 2.6-nm AuNPs). After induction of PC-3 tumors in mice and as soon as the tumors had reached a mean volume of approximately 200 mm^3^, mice were randomized according to their individual tumor volume into three groups of 67 mice (control IT injection, TiONts-DTX, and TiONts-AuNPs-PEG_3000_-DTX).

To evaluate the effectiveness of treatment, tumor growth was evaluated by the growth retardation parameter (time to reach a volume of 1000 mm^3^). The TV was recorded three times a week using calipers and calculated according to the following formula: TV = thickness × width × length × 0.5. Each group included six or seven mice; numbers were calculated considering inter-mouse variability. Tumor growth delay was compared between mice groups using the nonparametric Mann–Whitney test.

## 5. Conclusions

The elaborated nanohybrid (TiONts-AuNPs-PEG_3000_-DTX) was prepared using a step-by-step synthesis allowing for the precise characterization of each grafting step. A thorough characterization of the latter led to substantial results, showing the originality and innovation of the associations, particularly the AuNPs/TiONts combination. Grafting of gold nanoparticles, functionalized with DTDTPA, on the surface of TiONts-APTES was successful thanks to peptidic coupling. This pathway of grafting limited AuNP and TiONt agglomeration and ensured an even distribution of Au@DTDTPA NPs over the surface of TiONts. Moreover, gold nanoparticles provided nanohybrids with a remarkable colloidal stability under physiological conditions, improving in vitro and in vivo behavior for targeted biomedical applications. The significant amount of therapeutic agent (DTX) modified by PMPI on the TiONts surface showed that DTX-PMPI were covalently bound with thiol functions (from PEG_3000_ and DTDTPA) but also via free amine groups (APTES) depending on the pH. Thus, this study confirmed the potent therapeutic effect of our final nanohybrid after DTX grafting onto the surface of nanotubes. In vitro biological assays (MTS) highlighted the cytotoxic activity of DTX present on the surface of TiONts-AuNPs-PEG_3000_-DTX on human PC-3 prostate adenocarcinoma cells. Although nanohybrids’ cytotoxicity was lower than that of DTX alone (IC_50_ = 82 nM versus 3.1 nM, respectively), cytotoxic activity remained very high. These results proved a better access of TiONts-AuNPs-PEG_3000_-DTX to microtubules compared to first generation TiONts-DTX (without Au@DTDTPA NPs) (IC_50_ = 390 nM) [43], possibly suggesting a better internalization. This observation was not surprising, as new functionalized nanomaterial was better dispersed at physiological pH, and was thus more stable in suspension even though it exhibited a lower ζ-potential indicating a screening effect (the steric hindrance should prevail in this case). In addition, we successfully developed a safe nanocarrier of DTX to directly deliver this drug into prostate tumors (by IT injection), able to maintain it inside tumor cells for longer (at least 20 days, as demonstrated by biodistribution results in Balb/c nude mice), and to prevent its diffusion throughout the body, avoiding side effects. Therefore, the effectiveness of the selected therapeutic agent was improved. After combined IT injection with radiotherapy, TiONts-AuNPs-PEG_3000_-DTX nanohybrid improved treatment efficacy by delaying tumor growth compared to the homologous nanohybrids without gold nanoparticles. Gold can increase the effect already demonstrated for TiONts-DTX [49,52]. Finally, these functionalized TiONts appear as promising versatile tools in the biomedical field to fight cancer, prostate cancer in particular.

## Figures and Tables

**Figure 1 cancers-11-01962-f001:**
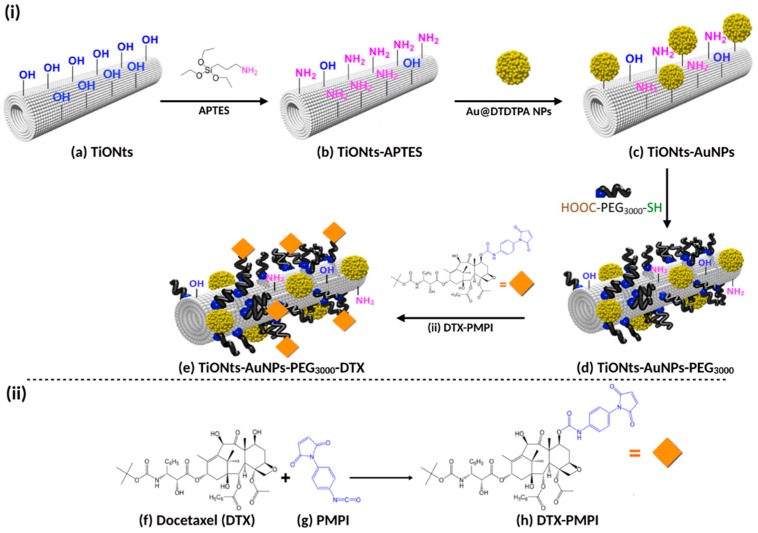
Illustration of (**i**) (**a**) titanate nanotubes (TiONts) and step-by-step pre-functionalization with (**b**) 3-aminopropyltriethoxysilane (APTES), (**c**) dithiolated diethylenetriaminepentaacetic acid-modified gold (Au@DTDTPA) nanoparticles (NPs), (**d**) α-acid,ω-thiol-polyethylene glycol (HS-PEG_3000_-COOH), and (**e**) *p*-maleimidophenyl isocyanate (PMPI)-modified docetaxel (DTX-PMPI) to yield the final nanohybrid—TiONts-AuNPs-PEG_3000_-DTX; (**ii**) this step corresponds to the modification of (**f**) DTX with (**g**) PMPI to form (**h**) DTX-PMPI (represented by an orange diamond).

**Figure 2 cancers-11-01962-f002:**
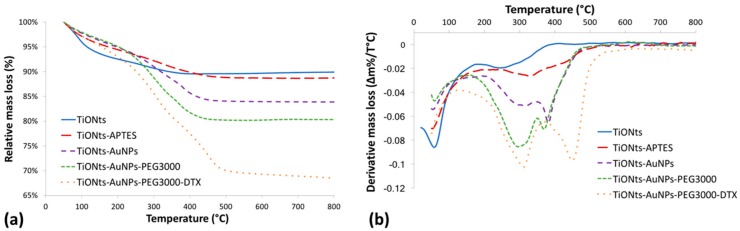
(**a**) Thermogravimetric analysis (TGA) and (**b**) derivative curves of bare TiONts and functionalized-TiONts under air atmosphere.

**Figure 3 cancers-11-01962-f003:**
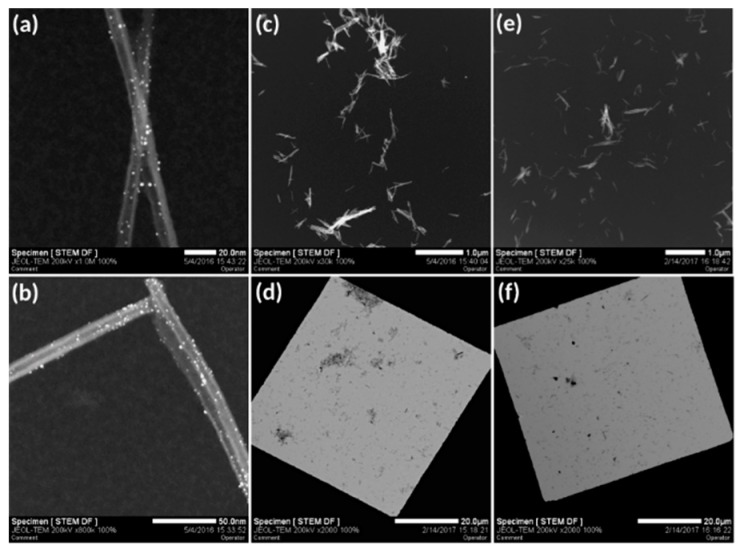
HAADF-STEM (high angle annular dark field scanning transmission electron microscopy) images showing (**a**,**b**) the grafting of Au@DTDTPA NPs on TiONts-APTES and the evolution of TiONts dispersion state (**c**,**d**) before (TiONts-AuNPs) and (**e**,**f**) after PEG_3000_ grafting (TiONts-AuNPs-PEG_3000_).

**Figure 4 cancers-11-01962-f004:**
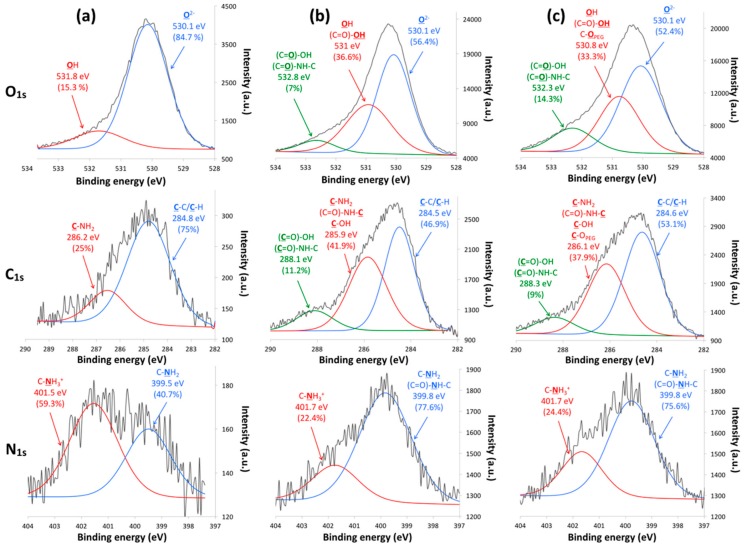
Fitted curves of O_1s_, C_1s_, and N_1s_ peaks in XPS spectra for (**a**) TiONts-APTES, (**b**) TiONts-AuNPs, and (**c**) TiONts-AuNPs-PEG_3000_.

**Figure 5 cancers-11-01962-f005:**
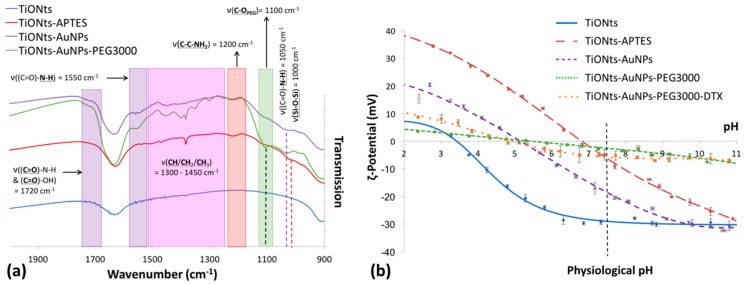
(**a**) FTIR spectra of TiONts, TiONts-APTES, TiONts-AuNPs, and TiONts-AuNPs-PEG_3000_ between 2000–900 cm^−1^ and (**b**) ζ-potential curves of bare TiONts and different functionalized TiONts (the vertical dashed line corresponds to the physiological pH).

**Figure 6 cancers-11-01962-f006:**
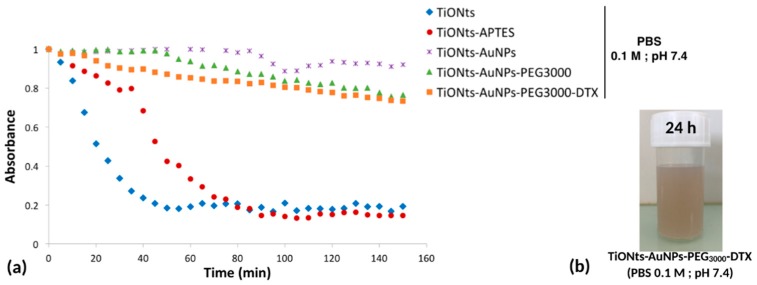
(**a**) Turbidimetric studies: colloidal stability of functionalized TiONt suspensions (phosphate buffered saline (PBS) 0.1 M; pH 7.4) over 150 min following their absorbance at 600 nm as a function of time. (**b**) Picture of a TiONts-AuNPs-PEG_3000_-DTX suspension in PBS (0.1 M; pH 7.4) after 24 h.

**Figure 7 cancers-11-01962-f007:**
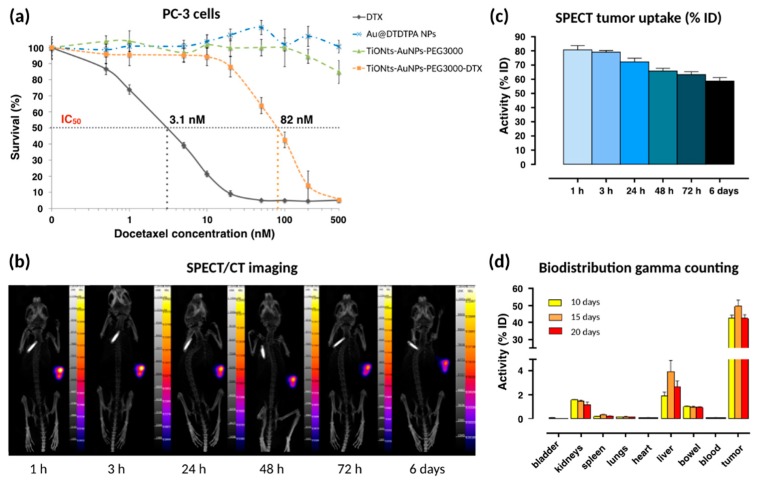
(**a**) Survival curves (3-(4,5-dimethylthiazol-2-yl)-5-(3-carboxymethoxyphenyl)-2-(4-ulfophenyl)-2H-tetrazolium (MTS) cytotoxicity assays) on PC-3 cell lines after incubation of DTX, Au@DTDTPA NPs, TiONts-AuNPs-PEG_3000_, and TiONts-AuNPs-PEG_3000_-DTX (mean ± SD). The studied range was from 0.5 to 500 nM in DTX concentration, which also corresponded to a concentration range of 4.1 × 10^−3^ to 4.1 μg.mL^−1^ for TiONts-AuNPs-PEG_3000_ and 3 × 10^−3^ to 3 μg.mL^−1^ for Au@DTDTPA NPs, present on the TiONts-AuNPs-PEG_3000_-DTX. The horizontal dotted line allows for an estimate of the different nanohybrids’ IC_50_. (**b**) Single photon emission computed tomography coupled with a conventional scanner (SPECT/CT) imaging of kinetics and (**c**) SPECT tumor uptake (mean value ± SD) achieved in Balb/c nude male mice after intratumoral (IT) injection of TiONts-AuNPs-PEG_3000_-DTX-^111^In at 1 h, 3 h, 24 h, 48 h, 72 h, and 6 days (as a function of ^111^In injected activity (5.7–8.7 MBq) and corrected by ^111^In radioactive decay). (**d**) TiONts-AuNPs-PEG_3000_-DTX-^111^In biodistribution in dissected organs (bladder, kidney, spleen, lung, heart, liver, bowel, blood, and tumor) by radioactivity detection using gamma counting 10, 15, and 20 days after IT injection (mean value ± SD).

**Figure 8 cancers-11-01962-f008:**
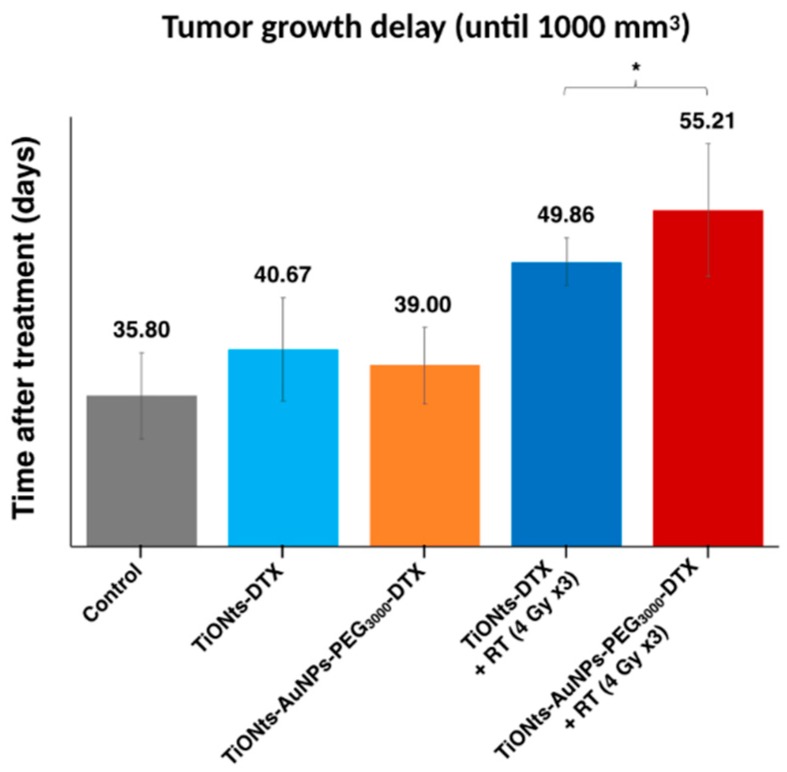
Therapeutic effect of control, TiONts-DTX, and TiONts-AuNPs-PEG_3000_-DTX, associated or not with radiotherapy (RT) administered with three daily fractions of 4 Gy (3 groups, *n* = 6–7), after IT injection into PC-3 xenografted tumors. * *p* = 0.035 (TiONts-AuNPs-PEG_3000_-DTX + RT vs. TiONts-DTX + RT); comparison performed using nonparametric Mann–Whitney test.

**Table 1 cancers-11-01962-t001:** Results of relative mass loss and graft ratio of bare TiONts and functionalized-TiONts.

Nanohybrid Name	Initial Temperature of Degradation (°C)	Relative Mass Loss (%)	Degraded Molecular Weight (g.mol^−1^)	Molecule.nm^−2^ (average)	Reproducibility (*n*)	Number of Grafted Molecules Per TiONt ^1^
TiONts	190	2.6	18	10.2 (±1.5) OH	10	−
TiONts-APTES	175	6.3	58	2.6 (±0.2) NH_2_	9	14,230
TiONts-AuNPs	150	12.4	511	0.40 (±0.05) DTDTPA	7	2,200
TiONts-AuNPs-PEG_3000_	150	16.1	3,073	0.040 (±0.003) PEG_3000_	4	220
TiONts-AuNPs-PEG_3000_-DTX	150	27.4	1,049	0.30 (±0.01) DTX-PMPI	2	1,700

^1^ The number of grafted molecules per TiONt was estimated by means of geometrical calculation considering only the external surface of TiONts.

**Table 2 cancers-11-01962-t002:** X-ray photoelectron spectroscopy (XPS) analyses: atomic concentration of bare TiONts, TiONts-APTES, TiONts-AuNPs, and TiONts-AuNPs-PEG_3000_.

Atomic Concentration (%)	C_1s_	O_1s_	Na_KLL_	Ti_2p_	N_1s_	Si_2p_	Au_4f_
TiONts	7.3	58.7	13.5	20.5	−	−	−
Elements (TiONts)/Ti (%)	0.3	2.9	0.7	1.0	−	−	−
TiONts-APTES	11.2	56.8	5.7	21.5	2.3	2.5	−
Elements (TiONts-APTES)/Ti (%)	0.5	2.6	0.3	1.0	0.1	0.1	−
TiONts-AuNPs	18.5	54.1	1.4	19.2	3.1	2.5	1.2
Elements (TiONts-AuNPs)/Ti (%)	1.0	2.8	0.1	1.0	0.2	0.1	0.1
TiONts-AuNPs-PEG_3000_	23.8	52.1	0.5	17.5	2.9	2.0	1.2
Elements (TiONts-AuNPs-PEG_3000_)/Ti (%)	1.4	3.0	≈ 0	1.0	0.2	0.1	0.1

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
