# Peer review of "Titanate Nanotubes Engineered with Gold Nanoparticles and Docetaxel to Enhance Radiotherapy on Xenografted Prostate Tumors"

_cancers, 2019, doi:10.3390/cancers11121962_

Round 1
Reviewer 1 Report
Authors present the study of Radiotherapy Enhancement in Prostate Cancer Treatment by Titanate Nanotubes Engineered with Gold Nanoparticles and Docetaxel. They present the detailed description of the synthesis of complex nanoparticle however just few experiments are dedicated to its preclinical validation. The following attention shall be brought to authors’attention.
1.) Title: Since the presented experiments are mostly dedicated to process of synthesis (5 figures) of very complex nanoparticle and just 2 figures are dedicated to proper evaluation of the benefits of the treatment efficacy, I would suggest to modify the title of the article according its real achievements.
2.) Introduction: The introduction (line 49) mention epidemiology of prostate cancer in US but regarding the origin of authors and the journal, I would suggest to add the data regarding EU and/or world as well
3.) My biggest concern is related with part of introduction describing the benefits of nanoparticles to the standard cancer treatment. The authors describe the effect of bio-distribution after i.v. injection, mechanism of EPR effect linked with i.v. injection (line 65), good biodistribution of AU-NPs after i.v. injection and its renal clearance that may be overcome with IoONts (121-127 line) etc., however in the article the authors use just i.t. injection. Thus, in my opinion, the introduction is not coherent with the experimental design and the achieved results.
4.) Results: In the results section authors describe that the nano-platform can be use for nuclear imaging and therapy. Which results support the possibility of imaging of TiONts-AuNPs-PEG3000-DTX. If it is just a supposition than this statement should be presented with less strength.
5.)What is the structure and dimensions of final nanoparticle including DTX? Could you add TEM images?
6.) Table 1. Explain the reproducibility column (n). Why the n of AuNPs-PEG3000-DTX is so low (2)?
7.) Figure 3. Include HADF-STEM image of AuNPs-PEG3000-DTX or an explanation why it is not included.
8.) Table 2. Include XPS analysis of AuNPs-PEG3000-DTX or an explanation why it is not included.
9.) Figure 4. Include data of AuNPs-PEG3000-DTX or an explanation why it is not included.
10.) Figure 5a. Include FTIR spectra of AuNPs-PEG3000-DTX or an explanation why it is not included.
11.) Figure 6. To prove the colloidal stability under physiological condition, the serum media should be examined.
12.) Figure 7a. More prostate cell lines should be tested and the decrease of therapeutic activity of Docetaxel bound to nanoparticles should be discussed. For that the results of loading and release of the drug should be presented
13.) Figure 7b. The biodistribution of nanoparticles after i.v. injection should be explored.
14.) Figure 8. Regarding the current title of the article, this should be the most important figure of this manuscript, however very important controls are missing: i.) RT alone, ii.) AuNPs (IT) + RT, iii.) DTX (i.t.) alone. Also the curves reflecting the tumor growth should be included.
15.) Conclusions:The presented conclusion could not be stated based on the obtained results and should be completely reformulated.
Line 554: not all grafting steps were precisely characterized. Data concerning TiONts-AuNPs-PEG3000-DTX are missing in several occasions (describe above)
Line 560: Just the results of the stability in PBS are presented. In the physiological conditions the nanoparticles will be in the blood that contains serum, various proteins and other molecules not in PBS. The authors do not present the results confirming stability in vivo.
Line 565: The results shows considerably less efficacy of the TiONts-AuNPs-PEG3000-DTX compared to free DTX thus it is not adequate to speak about potent therapeutic effect
Line 570: The comparison of efficacy in vitro of TiONts-AuNPs-PEG3000-DTX with TiONts- DTX is just theoretical and should not be stated as an conclusion without the experimental validation (cytotoxicity analysis provided in parallel single experiment)
Line 570: Further conclusion regarding internalization, without any experimental data, are just a supposition and should be experimentally proved.
Line 575: If the TiONts-AuNPs-PEG3000-DTX platform is safe, why it was not contrasted with free DTX i.t. of injected i.v.?
Line 577: The authors state that the efficacy of therapeutic agent (DTX) is improved without even use the free drug in the experiment investigating the efficacy.
Reviewer 2 Report
This is a nice manuscript that develops and tests a multi-functional chemo-radiosensitising nano platform for cancer therapy.
However, there are a few modifications that are required before I can recommend publication. These are a mixture of minor and major comments that I will deal with in the order that they appear in the manuscript.
1) Line 87, minor point. Would prefer 'compared with' instead of 'on the contrary of'.
2) Line 123, minor point. Would prefer 'the intravenous injection and injection of'
3) Line 130, minor point. Would prefer 'next generation nanohybrid'
4) Figure 1(i). Is there any possibility in step c-->d that the thiol group of PEG3000 will compete with DTDTPA for attachment to AuNPs? If so, this would give a different structure to the overall NP that could affect bioavailability and stability.
5) Figure 1(ii). Please clarify whether the orange diamond represents CTX-PMPI or some leaving group. If it's DTX-PMPI, then an equals sign, rather than an arrow might be more appropriate.
6) Figure 3 c-f are not explained in the text.
7) Lines 193, 226, 230. Avoid using the word 'proves' or 'proving'. Instead use something like 'consistent with'.
8) Similarly in line 197, would prefer 'consistent with' instead of 'demonstrating'.
9) Line 246 minor point. Would prefer 'physiological pH'.
10) Line 250. Not sure what is meant by ...'density grafted less important'...
11) Line 283 and Figure 6. Turbidimetric analysis not explained in the Methods section. How was this done? Was the absorbance at any specific wavelength?
12) Lines 298-303 and Figure 7a. I think it's misleading to use Docetaxel concentration on the x-axis of the graph, especially when there is no DTX present in two of the nanohybrids tested. Can these data be re-plotted in terms of the molarity of each nanohybrid? Also, the y-axis of the graph should be 'survival %'.
13) Lines 310 and 570-571. Difficult to talk about improved cell internalisation without ICPMS of treated cells to quantify cell accumulation (either on or in the cells). This information would make the paper stronger.
14) Lines 320-326. Radiolabel detection/imaging does not prove the presence of intact nanohybrids in cells over time. Potentially, iodine could be lost or exchanged on the DTDTPA, giving a misleading view of where the NP is located. It would be good to mention this caveat. In fact, this aspect could be tested if you have any data looking at the thyroid for evidence of free iodine accumulation.
15) Figure 8. This experiment has a control - RT, but no control + RT. Need to include a control+RT. Also, it would be more informative to re-plot these data as tumour volume versus time, using a series of line graphs.
Reviewer 3 Report
With interest I reviewed the manuscript submitted by Loiseau et al. They aimed to develop a new generation of titanate nanotubes (TiONts) to enhance radiotherapy in prostate cancer. The surface of TiONts was coated with a siloxane and coupled with DTDTPA-modified gold nanoparticles and a heterobifunctional polymer to improve suspension stability. docetaxel was coupled as a therapeutic agent. The in vitro studies performed could show that this TiONts have a distinct cytotoxic activity on human PC-3 prostate cancer cells. In PC-3 xenografted tumors on mice, the TiONts prepared as mentioned above were retained within the tumor after intratumoral injection allowing to delay tumor growth after irradiation. Accordingly, this work provides an important basis for further cancer research.
In general, the manuscript is concise, very well written and the methods are sound.
However, some minor issues have to be considered:
- The abbreviations used must be introduced consequently from the beginning or already in the introduction section (e.g., STEM, XPS, etc.)
- As suggested by the author guidelines of Cancers, the work should be rewritten. The results and the discussion should be separated from each other. I.e., the discussion should be conducted separately in a discussion section.
Reviewer 4 Report
The manuscript entitled " Radiotherapy Enhancement in Prostate Cancer Treatment by Titanate Nanotubes Engineered with Gold Nanoparticles and Docetaxel " written by Alexis Loiseau and co-authors reports on synthesis and development of novel nano-engineered nanohybrids starting from Titanate Nanotubes, coated with APTES and conjugated with DTDTPA-modified Au nanoparticles and PEG. This hybrid complex system was characterized by several techniques and used for in vivo Bioimaging and and in-vitro investigations. Authors showed a radiotherapy enhancement by testing them against Prostate Cancers. The paper is well structured and it is surely suitable for Cancers readers but is not yet ready to be published in this form, since it needs substantial revisions in order to reach standard level for Cancers.
A number of issues that authors need to ammend\address are listed in the following part:
1) In the introduction authors have overlooked a possible comparision with a competitive type of nanotubes for drug delivery and bioimaging: halloysite clay nanotubes (HNTs). Please revise it and after comparing them with Titanate NTBs
2) What about uptake and internalisation study for NTBs inside prostate cancer cells? (eg FACS or CLSM study are suggested to be performed accordingly)
3) Release study are also missed. Please update\address this point.
4) What is the rationale in cancer cells choice? A thoroughly discussion about motivations\reasons of it is recommended.
5) For in vivo data it would be good to report tomour volumes variation pictures upon drug loaded-NTBs treatment (what about ex vivo data?)
6) It would be important to discuss\report what is the fate of this NTBs after treatment (eg bioddegradability is an important issue). Please address this issue.
7) A moderate English revision by native speaker would also improve the text comprehension and the readability of manuscript itself
8) Finally, a comprehensive revision\update of literature references is also suggested after revision (eg upon updating HNTs contribution and thereby references)
Round 2
Reviewer 1 Report
The manuscript improved after its revision significantly. Authors reply and/or corrected all the queries and the manuscript is now ready to be published.
Reviewer 2 Report
This manuscript is now improved and suitable for publication
Reviewer 4 Report
Authors have properly addressed all major concerns\issues raised by previous report, thereby the manuscript can be accepted in present (revised) form